# Chromatin Architectural Factors as Safeguards against Excessive Supercoiling during DNA Replication

**DOI:** 10.3390/ijms21124504

**Published:** 2020-06-24

**Authors:** Syed Moiz Ahmed, Peter Dröge

**Affiliations:** School of Biological Sciences, Nanyang Technological University, Singapore 637551, Singapore; syed0051@e.ntu.edu.sg

**Keywords:** DNA topological strain, DNA/chromatin supercoiling, topoisomerases, chromatin architectural factors, replication stress, GapR, HMGA2

## Abstract

Key DNA transactions, such as genome replication and transcription, rely on the speedy translocation of specialized protein complexes along a double-stranded, right-handed helical template. Physical tethering of these molecular machines during translocation, in conjunction with their internal architectural features, generates DNA topological strain in the form of template supercoiling. It is known that the build-up of transient excessive supercoiling poses severe threats to genome function and stability and that highly specialized enzymes—the topoisomerases (TOP)—have evolved to mitigate these threats. Furthermore, due to their intracellular abundance and fast supercoil relaxation rates, it is generally assumed that these enzymes are sufficient in coping with genome-wide bursts of excessive supercoiling. However, the recent discoveries of chromatin architectural factors that play important accessory functions have cast reasonable doubts on this concept. Here, we reviewed the background of these new findings and described emerging models of how these accessory factors contribute to supercoil homeostasis. We focused on DNA replication and the generation of positive (+) supercoiling in front of replisomes, where two accessory factors—GapR and HMGA2—from pro- and eukaryotic cells, respectively, appear to play important roles as sinks for excessive (+) supercoiling by employing a combination of supercoil constrainment and activation of topoisomerases. Looking forward, we expect that additional factors will be identified in the future as part of an expanding cellular repertoire to cope with bursts of topological strain. Furthermore, identifying antagonists that target these accessory factors and work synergistically with clinically relevant topoisomerase inhibitors could become an interesting novel strategy, leading to improved treatment outcomes.

## 1. Introduction

The vast majority of DNA-related processes are executed by highly specialized nucleoprotein complexes [1] and occurs over a wide range of time scales [2]. Human genome replication, for example, is a dynamic and complex DNA transaction that spans several hours during S-phase and takes place simultaneously at high speed at many genomic locations. At the same time, replication must be tightly monitored to preserve genome integrity and genetic information for the progeny cells [3,4]. 

Any impediment to the replication machinery that causes DNA synthesis to prematurely pause or stall permanently will induce replication stress [5,6]. Most of the real-life impediments have been identified and studied in detail. They include deprivation of precursors [7,8], alternate DNA secondary structures in the template DNA [9,10], deregulated origin firing [11,12], and collision of transcription-replication complexes [13,14] to name but a few. However, what appears to be missing, in our opinion, is a wider recognition of localized DNA topological strain in the form of excessive DNA/chromatin fiber windings about itself, also known as DNA/chromatin supercoiling, as another prominent cause of replication stress and a threat to genome stability. Although progress in this field has been made, such lack of recognition is probably due to the fact that experimental interrogation of individual dynamic supercoiling processes on sub-second timescales in situ remains technically challenging and will require the development of novel experimental approaches [15,16].

With this brief review, we hoped to bring this fascinating topic closer to the attention of a broader scientific audience. We would argue that the extent of localized DNA/chromatin supercoiling sometimes threatens to exceed the capacity of the main cellular factors—the topoisomerase enzymes—to deal with this problem and that recently identified chromatin structural factors adopt unexpected and important accessory roles. For the sake of brevity, we limited our discussion to one of the major DNA transactions inside cells—genome replication. However, it should be emphasized that similar considerations apply to other highly dynamic transactions, such as transcription and homologous recombination, which can lead to interdependent DNA topological strain situations [6,14].

## 2. DNA Topological Ramifications during Replication

The large protein assemblies responsible for carrying out the complex task of eukaryotic DNA replication, termed replisomes, translocate along the two intertwined template strands of the chromatinized DNA double helix at velocities of about 30 base pairs per second [17,18]. These highly sophisticated, fast traveling molecular machines are often physically tethered to protein factors or other replisomes, thus potentially forming so-called replication factories [19,20,21]. An important physical consequence of tethering is that replisomes are no longer free to rotate and, therefore, unable to follow the right-handed helical path of the parental template strands during translocation. Furthermore, the multitude of protein–protein interactions within replisomes severely restrict rotations of DNA polymerases [22], and DNA topology [23] predicts that this lack of rotational freedom in combination with the helicase-mediated disruption of base pairing in the template will reduce the twist parameter in the traversed parental DNA molecule without a concomitant change in the topological linkage of the two strands (i.e., the linking number; for a more detailed description see, for example, Yu and Droge [22]). The resulting imbalance between twist and linkage parameters leads to a highly dynamic scenario in which for each 10 base-pairs that a replisome progresses on the template, one positive (+) chromatin supercoil is theoretically generated in front of the replisome (Figure 1A). Based on a recent comprehensive 4D visualization study, up to ten thousand human replication forks are active in parallel during S-phase [17]. Combined with the replication speed of an individual fork in chromatin, one can estimate that tens of thousands of (+) supercoils are generated every second inside a replicating cell‘s nucleus, even if a fraction of replisomes remains free to rotate during translocation.

Rotation of the parental double helix about its own axis in front of a moving replisome would promote (+) supercoil diffusion, which prevents a potentially detrimental build-up of excessive torsional tension in the molecule. In order for diffusion to be effective, waves of (+) supercoiling must be able to traverse quickly and reach the end of the linear DNA template within a long chromatid fiber. However, because chromatin within eukaryotic chromosomes is often physically anchored to nuclear scaffolds or lamina [24] and also organized into topologically associated domains, supercoil diffusion is, in fact, severely compromised. Furthermore, the DNA ends of our linear chromosomes are organized into specialized nucleoprotein capping structures—the telomeres—which are considered as topological domain boundaries due to terminal DNA loop formation that prevents free rotation of the 3’ and 5’ ends about each other [25]. The large size of chromosomes and the rotational drag that accompanies large protein–DNA complexes, such as enhanceosomes, as well as compact heterochromatin structures, represent additional barriers to effective supercoil dissipation along chromosomal DNA [26,27]. Intriguingly, structural maintenance of chromosomes (SMC) cohesin complexes has also been recently identified as diffusion barriers, leading to a build-up of topological strain during S-phase in yeast, in particular, near centromeric regions and ribosomal RNA gene clusters [28]. Hence, based on a large body of evidence, localized waves of (+) chromatin supercoiling will inevitably occur in front of many translocating replisomes during S-phase, with more than ten thousand chromatin fiber rotations occurring every second over several hours.

Given the limited diffusion potential for supercoils within chromatids, an interesting open question is whether the resulting build-up of dynamic waves of (+) supercoiling generated in front of replisomes can always be effectively controlled by cellular factors, so that fork progression continues unhindered. In the following, we have first discussed some of the major consequences that excessive (+) topological strain can exert on replication forks, before turning our attention to the current understanding of cellular strategies to prevent its build-up.

## 3. Threats Imposed by DNA Topological Strain to Stalled Replication Forks

The build-up of excessive topological strain in the form of (+) supercoiling in front of a translocating replisome will induce fork stalling and cessation of DNA synthesis (Figure 1A) [29]. Various scenarios can unfold individually or concurrently if high (+) torsional strain in the parental DNA is not immediately removed. For example, the presence of (+) supercoiling in a chromatin molecule is known to favor nucleosome disassembly, thereby potentially altering the local chromatin structure [30]. It is unknown, however, whether this temporal structural alteration has in vivo consequences for genome stability and gene expression, or whether the biophysical properties of chromatin fibers constitute a topological buffer to accommodate torsional strain, as has been suggested based on single-molecule studies [31]. Furthermore, translocating transcription machineries are often tethered through gene gating. Similar to the scenario with anchored replisomes, Liu and Wang [32] were the first to propose that this immobilization would generate waves of (+) supercoiling ahead and negative (-) supercoiling behind a translocating RNA polymerase, respectively [33]. When replication and transcription machineries converge, the temporal build-up of high localized topological strain could reach very high levels, leading to the stalling of both replication and transcription complexes [14]. Due to the high torsional strain, DNA topology predicts two outcomes for stalled forks [34].

First, precatenane structures can form on the two newly replicated daughter chromatin fibers behind the replication fork. Precatenanes are defined by multiple intertwinings of double-stranded sister chromatin segments, thus forming a braided chromatin structure (Figure 1A) [35]. Their formation is energetically driven by the torque of (+) supercoiled DNA, which propels the excessive topological linkage in the parental DNA to a position behind the fork. This process requires rotation of the replisome, i.e., any tether must first be untied before precatenane nodes can form [36]. Unresolved precatenanes will eventually lead to catenated sister chromatids at the end of the S-phase, which, in turn, poses a threat to genome stability during mitosis. Furthermore, there is evidence that due to sterical hindrance, precatenane formation might interfere with lagging strand DNA synthesis. This would result in extended regions of single-stranded DNA and potentially trigger genome instability due to endonucleolytic attack [37,38]. However, recent data indicated the existence of physical chromatin features that could intrinsically limit precatenane formation. Using direct torque measurements, Wang and colleagues [39] demonstrated that whereas a single chromatin fiber was torsionally soft, a braided precatenane fiber was much stiffer. This led to the proposition that the topological strain at replication forks is preferentially directed in front of the fork, i.e., in the form of (+) chromatin supercoiling. 

A second, highly contested topology-driven consequence at stalled forks is a process called fork regression, also known as fork reversal [40,41]. This remodeling process generates a four-way DNA assembly that structurally resembles a Holliday junction (HJ) (Figure 1B). In this case, the torque of (+) supercoiling “pushes” the stalled branch point of the fork backward, with re-annealing of base pairs and concomitant re-winding of parental strands. DNA topology predicts that this re-introduction of twist in the parental DNA molecule relaxes the (+) supercoiling. Reversal into HJ structures also requires the melting of base pairing within the two nascent daughter duplexes. Due to strand complementarity, this process can lead to the formation of a new fourth duplex (Figure 1B).

Regressed fork structures have been defamed as pathological, in part because their formation involves massive replisome disassembly that could induce complete fork collapse and genome instability. Regressed forks are prone to cleavage by endonucleases called HJ resolvases, such as MRE11 or EXO1 [42]. However, there is compelling evidence that the formation of these structures can also be actively promoted by specialized cellular factors and become important intermediates in homologous recombination-mediated re-start of stalled replication forks, thereby contributing to genome stability under conditions of replication stress [43,44]. We think it is possible that in situations where many stalled forks simultaneously experience excessive topological strain, the cellular capacity to deal with these forks may become exhausted, and the balance between these two outcomes may shift towards pathological regressed forks and trigger fork collapse, leading to genome instability or apoptosis. Hence, given that the topological threats to stalled replication forks are potentially detrimental to genome stability, we have next briefly summarized the established solutions that have evolved to minimize these threats.

## 4. Established Solutions at Topologically Stressed Replication Forks

Enzymes called topoisomerases are found in all cellular life forms and serve as the main solutions to the topological “DNA linkage problem” [45,46,47]. A number of excellent reviews composed by founders and leaders in this field provide comprehensive and detailed views on the mechanisms of topoisomerase action and how these fascinating enzymes deal with topological ramifications, such as supercoiling, DNA pre/catenanes, and knots, to curb topological strain inside nuclei and organelles [48,49,50]. We have, therefore, limited our discussion here to localized topoisomerase actions at replication forks and, in particular, to activities that take place in front of replisomes where excessive (+) supercoiling nucleates. This remains an interesting topic, firstly, because, inside the densely populated chromatin space of the nucleoplasm, the kinetics of topoisomerase-binding to waves of (+) supercoiling are not well understood. Second, the conformations that a transiently (+) supercoiled chromatin fiber generated by a translocating replisome may adopt remain elusive [51]. Third, topoisomerase action always requires DNA strand breakage, which must occur either at some distance ahead of an approaching replisome in order to avoid catastrophic collisions, or demands replisome pausing during topoisomerase action. It follows that topoisomerase catalytic cycles must stop in time to complete strand ligation before the replisome arrives or resumes replication, respectively. In other words, solutions either engage tight coordination between replisomes and topoisomerases, or topoisomerase actions take place at a safe distance from replisomes/helicases in order to maintain genome stability during replication. Detailed knowledge of these dynamic scenarios is still lacking. In this context, targeting topoisomerases with chemotherapy is of high clinical importance but often lacks sufficient efficacy and, hence, could be further improved by deeper understandings of the molecular mechanisms dealing with excessive localized topological strain [52].

### 4.1. Actions of Eukaryotic Topoisomerases Downstream of Replication Forks

Two types of topoisomerases have been classified primarily based on their mode of DNA strand cleavage: topoisomerase type 1 enzymes (TOP1) transiently cleave one DNA strand of a duplex segment at a time, while topoisomerase type 2 enzymes (TOP2) introduce a transient DNA double-strand break into the supercoiled substrate [47]. Although fundamentally different mechanisms are employed, both types of enzymes ultimately perform controlled DNA strand passage reactions, which change the topological linking number in the DNA substrate. This linking number change occurs in steps of 1 for TOP1 and steps of 2 for TOP2. In general, DNA strand passage is followed by re-ligation of the lesion to complete the reaction cycle. By acting on a wave of (+) supercoiled DNA in front of a replisome, strand passage by TOP1 or TOP2 will thus relax one or two (+) supercoils at a time within the chromatin fiber, respectively. Relaxation not only prevents the build-up of (+) supercoiling to facilitating replication fork progression but is also a key to fulfilling the strict requirement that the linking number of the two DNA strands in the parental human genome (about 2.3 x 10^8^) has to be reduced to zero during replication for the two daughter genomes to segregate into progeny cells at the end of mitosis.

The human genome contains at least six genes coding for different topoisomerases [48,50]. At the eukaryotic replication fork, TOP1B is primarily responsible for relaxation of (+) supercoils by permitting controlled rotations of one of the cleaved DNA ends around the remaining intact single strand, thus fulfilling the initially proposed role of a “swivelase” activity [53]. But how does TOP1B locate a moving and transient wave of (+) supercoiling in front of a fork within a reasonably short time frame, and, once associated with its substrate, how efficiently can the enzyme relax (+) supercoiling compared to supercoil generation?

First, the search mode can be diffusion-controlled and hence depends on a number of parameters, including variations in local enzyme concentrations within the nucleoplasm (Figure 2A). The copy number of TOP1B enzymes is estimated to be between 2 × 10^4^ to 1 × 10^5^ copies per HeLa and rat liver cell [54,55], which would easily match the number of supercoils generated per second during S-phase. TOP1B finds its substrate in the form of double-stranded DNA crossings [56], which is a key feature of plectonemic supercoiled DNA [23]. However, whether transiently (+) supercoiled chromatin fibers adopt a similar interwound structure has not been established [51]. In any case, diffusion-controlled search modes would imply that a certain level of recognizable (+) chromatin supercoiling must have been generated by a replisome. This conclusion is supported by findings that TOP1B appears to be active within a 600-base-pair (bp) region spanning moving forks in yeast [57]. However, it is known that TOP1B also binds to and even cleaves non-supercoiled double-stranded DNA, in particular, at positions of wrongly incorporated ribonucleotides during replication, i.e., behind the replisome [58]. This might slow down diffusion-controlled searches for topologically stressed chromatin by reducing the number of non-engaged enzymes. In addition, other DNA tracking processes, in particular, transcription, also generate thousands of supercoils per second during S-phase genome-wide, which compete for TOP1B action.

Second, TOP1B might be recruited by chromatin factors to genomic loci, where topological strain is more likely to build-up during replication. There is evidence that this occurs at regions that harbor special topological barriers, such as ribosomal gene cluster replication fork obstacles in yeast [59] (Figure 2A).

Third, TOP1B could be associated and travel with components of the replisome. For example, the direct binding of TOP1B to the ‘facilitates chro matin transcription’ (FACT) protein complex has been reported [60]. FACT functions as histone chaperone during replication close to the replicative helicase [38]. Reconstitution experiments with yeast proteins have also indicated that direct physical interactions between TOP1 and replication complexes might occur (Figure 2A) [61]. In addition, the viral SV40 helicase large T-antigen that works together with the host cell replisome and replaces the human minichromosome maintenance protein complex (MCM) replicative helicase interacts with TOP1B and could guide the enzyme to a location where its action is immediately required [62]. However, in scenarios of direct physical linkage with replisome components, it is inconceivable that topoisomerase and helicase/replisome actions occur simultaneously in the parental DNA without risking catastrophic consequences, such as replication run-off, leading to genome instability. Hence, if functional TOP1B does indeed hitch a ride with a replisome, short cycles of replisome/helicase action and pausing may be a prerequisite to provide the enzyme with sufficient time windows to safely mitigate (+) supercoil build-up. Such an intricate stop-and-go approach seems to work for TOP1B in complex with the transcription machinery [63].

Fourth, efficient relaxation of (+) supercoiling depends on the speed of the reaction. This has been examined in vitro on purified supercoiled plasmids and DNA single-molecule assemblies [64,65] and is in its processive mode relatively fast, with about 100 supercoils per second [26]. Single-molecule experiments furthermore have revealed that the DNA relaxation rate correlates with the net torque in the substrate and remains similar irrespective of the direction of strand rotation, i.e., TOP1B does not seem to have an intrinsic preference to relax (+) and (-) DNA supercoils [65,66]. This property is preserved in chromatinized DNA. Such fast DNA relaxation rates would indicate that a single TOP1B molecule at a replication fork could cope with the topological strain generated by the replisome. However, it is important to note that the relaxation rate is about 10 times slower on chromatinized DNA than on naked DNA [67]. This reduction is probably due to the rotational drag on DNA imposed by nucleosomes and seems to be similar for (+) and (-) supercoiled chromatin [68].

In summary, it appears that TOP1B has evolved different strategies to attend to the problem of highly dynamic (+) topological strain during replication. Importantly, genetic studies in yeast have revealed that the complete absence of TOP1B activity only leads to minor replication problems, in particular, on longer chromosomes. It has been argued that the latter is due to the aforementioned limitations in supercoil dissipation along longer chromatin fibers [69]. This finding and numerous other studies have indicated that a different type of eukaryotic topoisomerase—TOP2A—is able to assist in fork progression [70,71]. Hence, TOP1B and TOP2A seem to generate a strong level of functional redundancy in (+) supercoiling relaxation, and replication ceases in the absence of both enzymes [57,68,72]. 

TOP2A exhibits a marked preference for relaxation of (+) supercoiled episomal chromatin over (-) supercoiled domains in yeast and, like TOP1B, recognizes double-stranded DNA segment crossings as initial substrate docking sites [56,68]. Whether this preference for (+) supercoiling as the substrate is due to special features of the episomes investigated in this study or is maintained on natural supercoiled chromatin fibers remains unknown, however. It has been estimated that the number of TOP2 molecules is between 10^5^–10^6^ per HeLa and skin fibroblast cell [54,55,73], which would imply that within the nucleoplasm, TOP2 molecules are in excess of the number of (+) supercoils generated at any time point during S-phase (Figure 2B). Similar to TOP1B in yeast, TOP2A appears to be active within a 600-base-pair (bp) region around moving forks [57], and, in vitro, TOP2A is able to relax supercoils at a rate of about 3 per second [64]. Furthermore, TOP2A has been found in association with scaffold/matrix attachment regions (S/MARs; [74]) and appears to act functionally synergistic with the SMC cohesion complex that has recently been identified as a topological barrier to supercoil diffusion (Figure 2B) [75,76].

Taken together, these findings indicate that both types of eukaryotic topoisomerases are present in high copy numbers in the nucleoplasm, possess fast reaction rates, and can be strategically positioned to become activated once a wave of (+) chromatin supercoiling generated by moving replication forks builds up at certain genomic loci. It should be noted, however, that earlier in vitro studies using a model system that functionally links recombination to transcription via transient waves of DNA supercoiling have revealed that both eukaryotic enzymes are unable to achieve complete relaxation even when present at high molar excess over substrate DNA [77]. These findings and the fact that both enzymes functionally assist each other in vivo point again to the challenge of maintaining (+) supercoiling homeostasis in front of replisomes during replication.

### 4.2. Prokaryotic Topoisomerases Acting in Front of Replication Forks

In contrast to eukaryotic cells where both types of topoisomerases contribute to the relaxation of (+) chromatin supercoiling in front of an active replisome, only type 2 enzymes, namely, DNA gyrase and TOPIV, appear to carry out this function in prokaryotic cells [78]. Furthermore, the speed of an individual *Escherichia coli* replisome is up to 1000 bp/sec and thus 30 times faster than a eukaryotic replisome [79,80]. Hence, assuming that the two replication forks in a single cell are not free to rotate, there will be about 200 (+) DNA supercoils generated each second in the remaining parental section of the replicating chromosome. This number could double if a new round of replication begins under favorable growth conditions before completion of the previous round.

Given that the *E. coli* chromosome is circular, supercoil diffusion reaching the end of the chromosome is not an option to avoid build-up of topological strain and even aggravates the problem because the two forks move towards each other. Furthermore, prokaryotic chromosomes are organized into looped domains, which restrict efficient diffusion along the entire chromosome [81,82].

Of the two enzymes, gyrase seems to provide the main activity that relaxes (+) supercoils in front of forks at an in vitro rate of about 2 supercoils per second [83]. A recent elegant single-molecule imaging study of gyrase activity in situ has provided some important insights into this dynamic process [80]. First, on average, about 600 gyrase molecules are present in a single cell, with only about 300 active molecules bound to a chromosome. Second, it has been found that only 8–12 gyrase molecules are in the vicinity of an individual replication fork, which would enable relaxation of about 24 (+) supercoils per second (assuming that the relaxation rate for gyrase measured in vitro also applies in vivo). This could indicate a mainly diffusion-controlled search mode for gyrase to associate with mobile (+) supercoiled substrates (Figure 2B). Based on these considerations, gyrase would not be able to keep up with the rate of supercoil generation (100/sec) at the fork. It has been suggested [80] that additional gyrase molecules, for example, bound at some distance within a topological domain that is being replicated, will assist in relaxing (+) supercoils that diffuse away from the replisome. However, the availability of sufficient gyrase molecules may be compromised due to transcription-induced (+) supercoiling that needs to be controlled in other domains of the genome. Furthermore, gyrase plays an important role in maintaining a certain level of overall unconstrained (-) supercoiling across the various topologically closed domains of the genome, which is vital for most genome functions and diverts more gyrase molecules from replication forks [34].

A study by Reyes-Lamothe, Possoz [79] has proposed that the two sister replisomes are not physically connected to each other during replication and are rather mobile within the cell. Hence, it is probably safe to assume that replisomes experience some degree of rotational freedom and, as outlined above (Figure 1A), this would promote the formation of precatenanes behind a replisome, and that needs to be resolved. Although TOPIV exhibits processivity during relaxation of (+) supercoils at a rate of ≈ 2.4 s^−1^ [64], its main action is most likely required behind the fork to resolve these precatenanes [78]. Furthermore, the copy number of DNA-bound TOPIV molecules per cell in S-phase (about 30) is much lower than that of gyrase (about 300), and only five molecules are found near translocating replisomes [80]. It should also be noted that similar to the situation in eukaryotic cells, transcription processes could generate many additional bursts of (+) DNA supercoiling, which compete also for TOPIV action during the replication phase.

In summary, it remains a point of debate whether the combined actions of the respective topoisomerases in pro- and eukaryotic cells, perhaps in conjunction with certain biophysical features of eukaryotic chromatin that, to some extent, could temporally buffer topological strain, are entirely sufficient in coping with the genome-wide generation of excessive transient (+) supercoiling and protect the genome during S-phase. In fact, recent studies have revealed that cells have evolved additional measures to mitigate the topological threat to genome stability during replication, thus indicating that the established solutions described above may not always be sufficient.

## 5. Architectural Chromatin Factors Emerging as Topological Sinks 

A multitude of different non-histone chromatin architectural factors exists in pro- and eukaryotic cells. These factors often bind duplex DNA without nucleotide sequence specificity, and some of them alter the local DNA structure through, for example, bending or wrapping, while others contribute to higher-order chromatin organization by simultaneously binding to more than one DNA duplex segment *in cis* and/or *in trans*. We have highlighted here two such factors (one each from pro- and eukaryotic cells) because they have been recently found to play important accessory roles in curbing excessive topological strain during replication. We proposed that their modes of action represent examples of general strategies on how cells expanded their repertoire to mitigate threats imposed by localized excessive topological strain.

### 5.1. GapR: a Mobile Supercoil Sink during Bacterial Genome Replication

Prokaryotic genomes are organized in higher-order nucleoprotein structures called nucleoids, and several cell cycle-regulated nucleoid-associated proteins (NAPs), such as IHF and HU in *E. coli*, play key roles that often go beyond mere DNA architectural activities [81]. The GapR protein is such a NAP found in *Caulobacter crescentus,* where it compacts and organizes the bacterial genome. Beyond this architectural role, the protein is crucial for the control of DNA replication and cell division [84,85], and cells lacking GapR exhibit cell division defects and replication fork stalling [86,87]. Interestingly, DNA-bound GapR co-localizes with replication forks at positions between the origin of replication and termination sites [87]. Taken together, these initial findings have pointed at a direct involvement of GapR in fork progression.

GapR is a relatively small protein and forms both homo-dimers and homo-tetramers (dimers-of-dimers), with the latter appearing to be physiologically more relevant. A GapR monomer consists of two short and an extended N-terminal α helix, and four monomers assemble into a repeating tetrameric unit with a large central channel that facilitates association with duplex DNA. GapR appears to exhibit a binding preference for AT-rich DNA, and once bound, the tetrameric protein undergoes conformational changes through monomer rearrangements, enabling it to scan along the duplex. When GapR encounters (+) supercoiled (overtwisted) DNA, the central channel constricts, which significantly increases the protein‘s binding affinity for such torsionally-strained DNA [88].

A recent elegant study has been able to unravel how GapR functions at forks during replication [89]. First, GapR significantly stimulates relaxation of (+) supercoiled DNA by both gyrase and TOPIV (Figure 3). Since these two enzymes act on DNA topological strain in front of translocating replisomes, this finding strongly argues for a need of extra protein factors at this location to quickly remove transient waves of (+) supercoiling. Second, in vitro experiments that investigated changes in DNA topology have revealed that GapR forms a clamp around overtwisted DNA, which cannot occur with the standard B-form duplex, thereby constraining (+) supercoiled DNA. The latter finding is important because constrainment of (+) supercoiling in front of a fork may neutralize or reduce the mechanical force that would otherwise work on the replisome and trigger the formation of precatenanes or regressed forks (Figure 1A,B). Based on the available data, a model emerges in which GapR, probably due to a two-dimensional DNA scanning search mode, first localizes to (+) supercoiled DNA generated by an active replication fork and stimulates gyrase and/or TOPIV to more efficiently relax (+) supercoils. Concurrently, GapR also prevents precatenane formation and replication fork regression through (+) supercoil constrainment (Figure 3).

It remains to be determined whether GapR directly binds to and recruits these topoisomerases to the (+) supercoiled domain or somehow alters the structure of (+) supercoiled substrates in such a manner that it increases the processivity of these enzymes. In any case, GapR‘s role in curbing (+) supercoiling appears not to be restricted to replication since the enzyme is also associated with the 3’ region of many highly transcribed genes, where transcription-induced (+) supercoiling is expected to accumulate [89]. Such strategic positioning could also help to mitigate the dangerous build-up of high levels of (+) supercoiling when replication and transcription machineries approach each other head-on, as mentioned above for replication and gene gating in eukaryotic cells.

Finally, we note that GapR is conserved across the α-proteobacteria without apparent homologs in other bacterial species. A few homologs have been identified in bacteriophages, including DsbA from bacteriophage T4 and GapR^Cr30^ from *Caulobacter*-specific phage ΦCr30. GapR and homologs show very little sequence similarity to other well-characterized DNA-binding motifs [90,91], and only weak similarities with eukaryotic proteins have been identified in the PFAM database [91]. Hence, while the specific mode of action of GapR in curbing dangerous excessive (+) supercoiling during replication might be restricted to a certain branch of prokaryotes, we proposed that these findings indicate a more wide-spread requirement for such activities to minimize replication stress due to localized supercoiling.

### 5.2. HMGA2: A Supercoil Sink for Topologically Stressed Replication Forks

The high-mobility group AT-hook 2 (HMGA2) protein is a non-histone architectural chromatin factor in higher eukaryotes and is normally expressed only during early phases of embryonic/fetal development. The protein has pleiotropic functions in transcriptional regulation, DNA repair, and cellular senescence [92,93]. Genetic studies in a variety of species, including humans, have revealed an important phenotypic connection between the expression level of HMGA2 in stem cells and body size [94], indicating that the protein plays a role in controlling the number of cell divisions during organismal development. Importantly, *HMGA2* is aberrantly re-expressed in many malignant cell types and strongly associated with tumorigenesis/metastasis in the adult organism [95,96,97]. Hence, HMGA2 appears to play an important role during cell proliferation in stem cells and in transformed cells.

HMGA2 is a small, mostly unstructured protein, which harbors three independent DNA binding domains, called AT-hooks, and a C-terminal acidic tail (Figure 4). The hooks are nearly identical and preferentially bind to the minor groove of AT-rich duplexes, where they introduce weak DNA bending [92,93,98]. The first indications that HMGA2 may have a function during cell proliferation, specifically at replication forks, have resulted from hydroxyurea-induced stalling of forks and the demonstration that HMGA2 co-localized with forks [99]. This and related studies have concluded that the protein has served as a chaperone during replication stress that protects forks from collapse into genotoxic double-strand breaks (DSBs), thus reducing the occurrence of chromosomal aberrations and apoptosis [99,100,101].

These initial findings have spurred several in-depth biophysical and biochemical investigations into the mechanistic details of the proposed fork chaperone function. They have revealed that HMGA2 exhibits the highest DNA binding affinity to (+) and (-) supercoiled substrates. Furthermore, upon binding to plasmids, HMGA2 alters the conformation of supercoiled DNA by scrunching the superhelix into a more elongated conformation, where intertwined DNA segments come in closer proximity to each other [102]. Using single-molecule assays, HMGA2 has interfered with TOP1B-mediated relaxation of (+) supercoiled DNA. Relaxation rates in the absence of HMGA2 are fast, but increases 300-fold in the presence of HMGA2 [103]. These data have revealed that the protein, like GapR, effectively constrains supercoiled DNA. Furthermore, supercoil constrainment is dependent on the presence of at least two functional AT-hooks in HMGA2, which has been interpreted to indicate that an HMGA2 monomer might bridge different duplex segments within a supercoiled domain (Figure 4) [102,104]. In conclusion, these results have shown that HMGA2 is able to stably associate with (+) and (-) supercoiled DNA and alters its conformation by constraining supercoils within HMGA2-DNA complexes [101]. One has to keep in mind, however, the unresolved question of whether HMGA2 functions in a similar manner in supercoiled chromatin.

Ahmed and Dröge recently demonstrated that HMGA2 significantly enhanced relaxation of supercoiled DNA by human TOP2A in vitro and that this stimulation also relied on the presence of at least two functional AT-hooks per HMGA2 molecule [103]. It remains to be determined whether this catalytic enhancement is mediated by the direct physical association between HMGA2 and TOP2, or whether supercoil scrunching by HMGA2, which brings duplex segments within a supercoiled domain in closer proximity to each other, generating more favorable TOP2A substrates that lead to faster relaxation rates.

In conjunction with aforementioned biochemical and biophysical data, additional experiments employing human cell-based assays and inhibitors of TOP1B and TOP2 have led to the following model that summarizes HMGA2‘s role in curbing topological strain at challenged forks (Figure 4): Upon inhibition of cellular TOP activity or through other mechanisms, such as helicase uncoupling [106], high levels of (+) chromatin supercoiling can accumulate in front of a fork and result in fork stalling. In cells expressing HMGA2, the protein will readily form specific complexes with the (+) supercoiled domain, thereby enhancing TOP2A-mediated supercoil relaxation, which could partially counteract the cellular effects of TOP inhibition. More efficient relaxation in conjunction with (+) supercoil constrainment reduces the torque on the stalled fork, which, in turn, could minimize replisome disruption, fork regression, precatenane formation, and, hence, fork collapse into double-strand breaks during replication stress. In this scenario, HMGA2 plays an important role in genome stability during replication stress, in particular, in fast replicating cells, such as embryonic stem and cancer cells. It is known that these cells are prone to replication stress, even in the absence of exogenous challenges [107,108]. In this context, finding HMGA2 antagonists that work synergistically with and enhance the efficacy of topoisomerase inhibitors could become clinically relevant [104].

An interesting unsolved question that applies to both GapR and HMGA2 is how, mechanistically, chromatin factors that constrain DNA supercoils are able to enhance supercoil relaxation by topoisomerases. We currently favor the following scenario: High-affinity binding of these factors to (+) supercoiled DNA recruits the respective topoisomerase through either direct physical interaction or the presentation of a favorable DNA substrate conformation. Once, topoisomerases begin to relax supercoils, the DNA affinity of the accessory factors is substantially weakened, and they depart from these locations. Due to increased local concentration effects [109], the topoisomerases are able to more efficiently complete substrate relaxation. In this scenario, chromatin factors serve some kind of nucleation function for efficient (+) supercoil removal by topoisomerases.

Finally, we like to mention that human HMGA2‘s close cousin, HMGA1, is more ubiquitously expressed during organismal development and in the adult body [110,111]. Similar to HMGA2, HMGA1 has been shown to crosslink different DNA segments through intra- and intermolecular DNA binding modes, thereby creating unique DNA scaffolds, such as loops and supercoil-like crossings in linear DNA molecules. HMGA1, via its three AT hooks, can also bind supercoiled plectonemic DNA [112,113] and change the helical periodicity of DNA on the surface of nucleosomal core particles [114]. Interestingly, the protein has been found to colocalize with TOP2 at AT-rich S/MARs [115,116]. Hence, backed by substantial experimental evidence, it has been proposed that HMGA1 can play a similar role as HMGA2 at replication forks [99].

## 6. Other Factors Potentially Mitigating DNA Topological Strain

The two examples described above highlight the importance of accessory factors in controlling transient bursts of topological strain during replication in order to maintain genome stability and function. Combining chromatin organization with DNA topology control functions is Nature‘s elegant solution, and it is, therefore, likely that other factors evolved to contribute in a similar way. However, while several chromatin factors in prokaryotic cells demarcate topological domains inside the cell, most of these abundant proteins constrain only (-) supercoiled DNA, and only a small subset appears to stably bind (+) supercoiled DNA. For example, bacterial NAPs, such as HU, H-NS, and FIS, preferentially bind to (-) supercoiled DNA [117,118,119]. Furthermore, several NAPs, such as SeqA, and the larger chromosome structuring protein MukB physically interact with and enhance TOPIV supercoil relaxation activity and the resolution of catenane nodes, but they don’t seem to cooperate with gyrase that works in front of forks [120,121,122].

In eukaryotic cells, perhaps the strongest evidence for a role of other factors in curbing (+) supercoiling generated during DNA transactions comes from studies of the tumor suppressor p53. The protein binds to both (-) and (+) supercoiled DNA, stimulates TOP1B relaxation activity [123,124], and appears to be critical in maintaining genomic stability during replication by preventing topological strain between converging replication and transcription complexes [125]. How p53 accomplishes this particular genome guardian function mechanistically remains to be elucidated, however.

## 7. Conclusions

The double-helical structure of duplex DNA generates dynamic topological problems during DNA transactions that involve fast-tracking of DNA-bound protein complexes. This becomes particularly problematic in front of replication forks. So far, topoisomerases have generally been considered as the main solutions to this problem. However, recent findings with chromatin architectural proteins in pro- and eukaryotic cells have highlighted the requirement for accessory factors during normal and perturbed DNA replication, respectively, to promote genome stability. These factors seem to act by enhancing topoisomerase action where it is most needed during replication, i.e., at the transient (+) supercoiled chromatin domain in front of replication forks. While it has not been established whether these factors promote the recruitment of topoisomerases to these locations or generate more favorable DNA substrate conformations (or both), it is clear that they efficiently constrain (+) DNA supercoiling, which, in turn, transiently reduces the torque imposed by (+) supercoiling on nearby replisomes or other nucleoprotein complexes. Hence, the various actions performed by these accessory factors appear to substantially contribute to genome stability during replication. Topoisomerase inhibitors prominently target the processes of DNA replication/cell proliferation and play important therapeutic roles as antibiotics and anti-cancer drugs, identifying antagonists that target these accessory factors could improve treatment outcomes in the clinic.

## Figures and Tables

**Figure 1 ijms-21-04504-f001:**
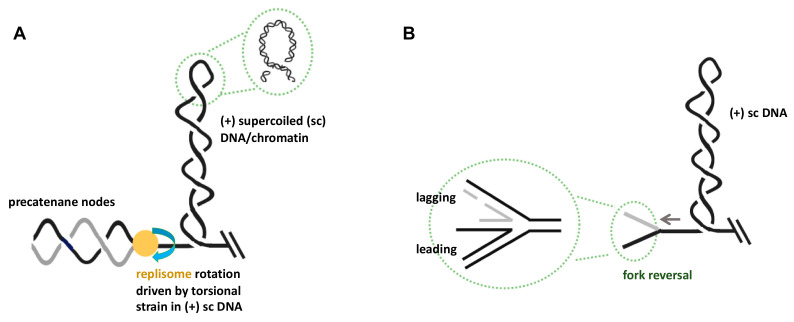
Consequences of excessive topological strain in the unreplicated DNA at stalled replication forks. (**A**) Topological strain in the form of left-handed interwound (plectonemic) duplex windings in the parental DNA (i.e., (+) supercoiling) can be transformed into right-handed braided precatenane structures behind the replisome. This transformation process will require the free rotation of replisomes or their disassembly. (**B**) Fork reversal as a consequence of topological strain ahead of the replication fork can lead to four-way DNA structures at the branch point. Note that for simplicity, chromatinization of DNA has been omitted in the drawings, and that duplex DNA is depicted as a single line. See text for detailed descriptions of these processes.

**Figure 2 ijms-21-04504-f002:**
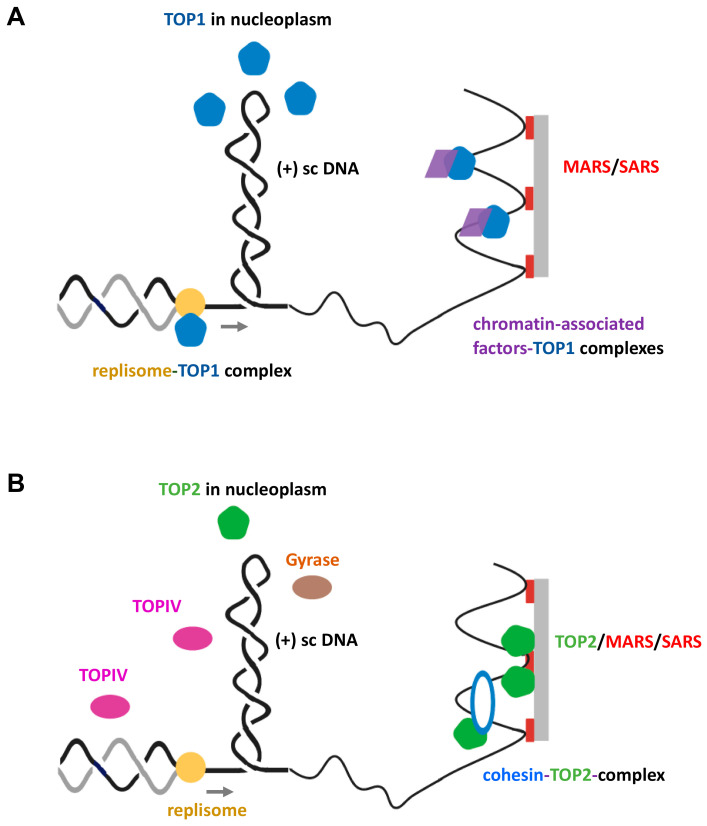
Various DNA topoisomerases function to maintain supercoil homeostasis at topologically stressed replication forks. (**A**) Topoisomerase type 1B (TOP1B) associates at regions of topological strain ahead of the replisome through diverse modes or travels in complex with the replisome, ultimately promoting DNA supercoil relaxation. (**B**) TOP2 enzymes preferentially relax (+) sc DNA ahead of the replication fork and can be physically associated with certain protein factors that are working as topological barriers throughout the genome. Note that TOPIV in prokaryotes is mainly responsible for the resolution of precatenanes behind the replisome.

**Figure 3 ijms-21-04504-f003:**
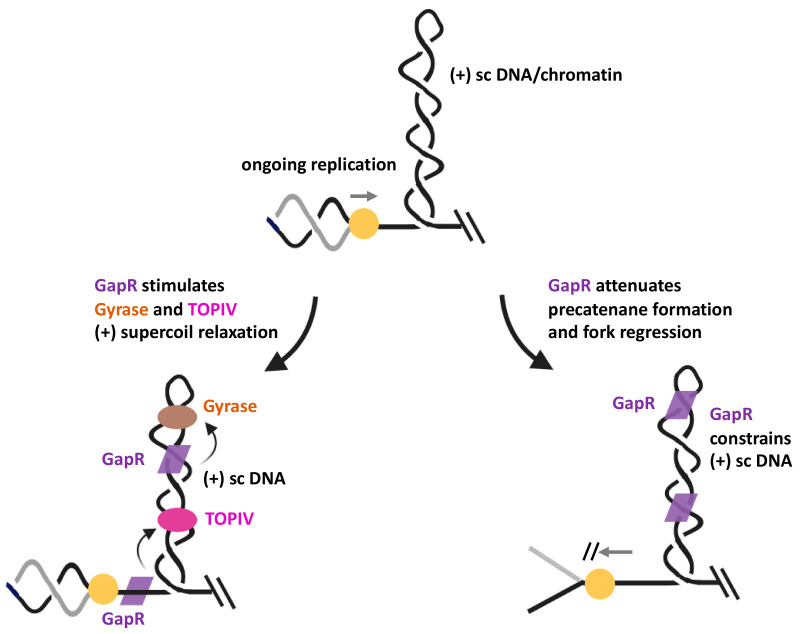
Proposed model for GapR functioning as a supercoil sink. GapR tracks along DNA and binds with a high affinity to overtwisted DNA, where it stimulates (+) supercoil relaxation by both bacterial type 2 topoisomerases (bottom left). The protein is also constraining (+) supercoils ahead of the replication fork, perhaps via some form of helix clamping [89], thereby potentially minimizing replication stress due to excessive formation of precatenanes or fork reversal (bottom right).

**Figure 4 ijms-21-04504-f004:**
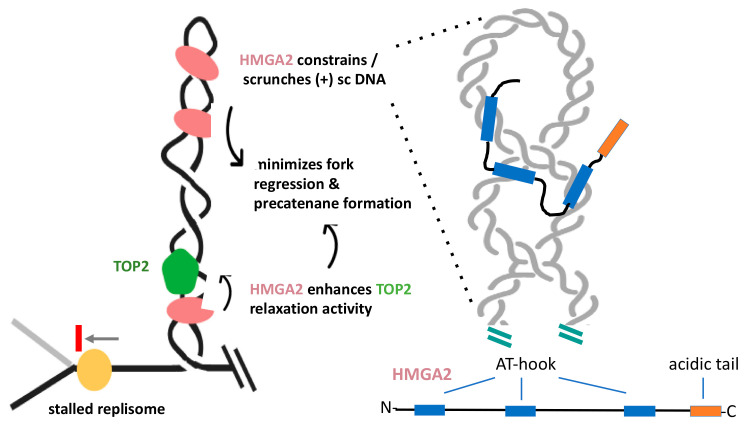
Proposed model depicting HMGA2’s role as a supercoil sink at topologically challenged replication forks. (**Left**) HMGA2 preferentially binds to and constrains supercoiled DNA and enhances TOP2-mediated supercoil relaxation. These activities stabilize the replisome and attenuate both precatenane formation and fork regression. (**Right**) Supercoil constrainment requires multiple AT-hook binding domains of HMGA2, which, based on results obtained with atomic force microscopy [102], bind to duplex segments at a DNA crossing in a plectoneme. This association will temporarily stabilize the supercoil and thus neutralize at least some of the torque experienced by the stalled replisome. Because HMGA2 appears to be able to form homodimers [105], it is possible that multiple HMGA2 molecules are working together at such a location and, furthermore, that this constrained DNA conformation represents a favorable substrate for TOP2.

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
