# Peer review of "Chromatin Architectural Factors as Safeguards against Excessive Supercoiling during DNA Replication"

_ijms, 2020, doi:10.3390/ijms21124504_

Round 1

Reviewer 1 Report

This review covers an important, often overlooked aspect, of chromatin structure and dynamics, as it is the dissolution and/or dissipation of DNA torsional stress. For the most part, the review is timely and appropriately written to grasp the attention of non specialists in the field.

However, some relevant facets should be corrected and improved:

- The quality of the drawings is very poor and the topology of DNA is ambiguous (if not wrong). Positively supercoiled plectonemes are not correctly traced in all 3 figures. Authors must correct DNA topology drawing and improve figure quality.

- Authors indicate the copy number and DNA relaxation rates of different topoisomerases to infer the limited capacity of these enzymes to reduce continuous DNA supercoiling waves during DNA replication. On that basis, the authors should also calculate or estimate (copy number and binding capacity to DNA crossovers) the actual capacity of GapR 20 and HMGA2 to constraint excess supercoiling.

- The review would benefit a lot by including figures of the structure of GapR 20 and HMGA2 complexes with DNA.

- Authors would like to explain (or speculate) how architectural proteins that stabilise DNA supercoils could at the same time facilitate supercoil removal, which seems contra-intuitive.

Author Response

This review covers an important, often overlooked aspect, of chromatin structure and dynamics, as it is the dissolution and/or dissipation of DNA torsional stress. For the most part, the review is timely and appropriately written to grasp the attention of non-specialists in the field.

However, some relevant facets should be corrected and improved:

Point 1: The quality of the drawings is very poor and the topology of DNA is ambiguous (if not wrong). Positively supercoiled plectonemes are not correctly traced in all 3 figures. Authors must correct DNA topology drawing and improve figure quality.

Response 1- We completely agree with this critical point and have now provided corresponding corrected diagrams and respective Figure Legends.

Point 2: Authors indicate the copy number and DNA relaxation rates of different topoisomerases to infer the limited capacity of these enzymes to reduce continuous DNA supercoiling waves during DNA replication. On that basis, the authors should also calculate or estimate (copy number and binding capacity to DNA crossovers) the actual capacity of GapR 20 and HMGA2 to constraint excess supercoiling.

Response 2- The reviewer raises an interesting point. However, after some deliberation, we think it might be too speculative to even estimate such capacities for GapR and HMGA2. Our main reasons are that at least for HMGA2, the copy number of molecules varies broadly between different cell types and tumor cells. For GapR it is estimated to be about 3000 (ref.88). Another confounding factor is that for both proteins, there is no real information about the number of molecules per supercoiled domain or DNA crossing. For GapR, for example, reference 88 states that there are likely multiple proteins involved in (+) supercoil constrainment. Hence, we think it would not add much substance to the review to include such vague estimates at the current state of knowledge.

Point 3: The review would benefit a lot by including figures of the structure of GapR 20 and HMGA2 complexes with DNA.

Response 3- We agree with this suggestion and have included a diagram depicting a model of HMGA2-scDNA complex describing supercoil constrainment by this factor in Figure 4. For GapR, however, the mode of supercoil constrainment is less clear. It could be either the clamping around overtwisted DNA or some form of higher order structure that involves the writhe of (+) supercoiled DNA. Hence, we think it might be too speculative to include a model diagram for GapR-DNA complex.

Point 4: Authors would like to explain (or speculate) how architectural proteins that stabilise DNA supercoils could at the same time facilitate supercoil removal, which seems contra-intuitive.

Response 4- We agree that this is a very interesting question and have included now a short paragraph describing our views on a possible mode of action on page 12, line 286.

Reviewer 2 Report

The review by S. M. Ahmed and P. Dröge is very interesting and well written. I appreciated the explanation how the replication induced positive supercoiling is relaxed by various topoisomerases and how proteins GapR and HMGA2 are implicated in the stimulation of topoisomerase-mediated relaxation of positive supercoiling. The described roles of GapR and HMGA2 proteins were mostly unknown to me.

I have though two critical comments:

1.In all figures the supercoiled plectoneme is wrongly presented. Positively supercoiled plectoneme should look like a left-handed superhelix. What authors presented is a kind of side-by-side model of supercoiling ,which would be physically and topologically wrong. Also, the precatenane winding in Fig. 1 is wrongly presented. The drawing shows some winding but mostly no winding as there is this side-by-side pattern. I understand that primary helix of DNA is not shown but winding of entire duplexes in plectonemes and in precatenanes needs to be presented correctly, especially in a review on DNA topology.

2.In the section 4.2. Prokaryotic Topoisomerases acting in front of Replication Forks the authors skipped the interesting problem that in prokaryotes one needs not only to relax positive supercoiling but also keep a significant level of negative supercoiling. That problem was treated at length in a recent review: Schvartzman et al., Closing the DNA replication cycle: from simple circular molecules to supercoiled and knotted DNA catenanes. Nucleic Acids Res 47, 7182–7198, 2019.  In my opinion, it would be good to quote that earlier review on action of DNA topoisomerases during DNA replications and also discuss the question how to relax positive supercoiling while keeping negative supercoiling.

Author Response

The review by S. M. Ahmed and P. Dröge is very interesting and well written. I appreciated the explanation how the replication induced positive supercoiling is relaxed by various topoisomerases and how proteins GapR and HMGA2 are implicated in the stimulation of topoisomerase-mediated relaxation of positive supercoiling. The described roles of GapR and HMGA2 proteins were mostly unknown to me.

I have though two critical comments:

- Point 1: In all figures the supercoiled plectoneme is wrongly presented. Positively supercoiled plectoneme should look like a left-handed superhelix. What authors presented is a kind of side-by-side model of supercoiling ,which would be physically and topologically wrong. Also, the precatenane winding in Fig. 1 is wrongly presented. The drawing shows some winding but mostly no winding as there is this side-by-side pattern. I understand that primary helix of DNA is not shown but winding of entire duplexes in plectonemes and in precatenanes needs to be presented correctly, especially in a review on DNA topology.

Response 1- As stated with respect to Reviewer 1, we could not agree more with this critical point and provide now corresponding corrected Figures and their Legends.

- Point 2: In the section 4.2. Prokaryotic Topoisomerases acting in front of Replication Forks the authors skipped the interesting problem that in prokaryotes one needs not only to relax positive supercoiling but also keep a significant level of negative supercoiling. That problem was treated at length in a recent review: Schvartzman et al., Closing the DNA replication cycle: from simple circular molecules to supercoiled and knotted DNA catenanes. Nucleic Acids Res 47, 7182–7198, 2019.  In my opinion, it would be good to quote that earlier review on action of DNA topoisomerases during DNA replications and also discuss the question how to relax positive supercoiling while keeping negative supercoiling.

Response 2- We are very grateful for reminding us about this inadvertent omission on our part. The mentioned NAR review by Schvartzman et al. is a very important contribution to this topic, and we have now included it as reference 34. It will greatly help the readers who look for background information on the topic of replication fork structures mentioned on page 3 of the revised manuscript. As suggested, we have also included a short new section on the necessity to maintain (-) supercoiling genome-wide in E.coli on page 8, beginning at line 328, in the context of available gyrase molecules to deal with (+) supercoiling.

Round 2

Reviewer 1 Report

I am satisfied with the responses and revision.